# RNAi-Mediated Functional Analysis Reveals the Regulation of Oocyte Vitellogenesis by Ecdysone Signaling in Two Coleoptera Species

**DOI:** 10.3390/biology12101284

**Published:** 2023-09-26

**Authors:** Xiaoqing Zhang, Lin Jin, Guoqing Li

**Affiliations:** Education Ministry Key Laboratory of Integrated Management of Crop Diseases and Pests, State & Local Joint Engineering Research Center of Green Pesticide Invention and Application, Department of Entomology, College of Plant Protection, Nanjing Agricultural University, Nanjing 210095, China; 2019202030@njau.edu.cn (X.Z.); jinlin@njau.edu.cn (L.J.)

**Keywords:** *Leptinotarsa decemlineata*, *Henosepilachna vigintioctopunctata*, ecdysone receptor, ultraspiracle, vitellogenesis, oocyte

## Abstract

**Simple Summary:**

In Coleoptera, successful female reproduction partially relies on effective vitellogenesis, characterized by vitellogenin (Vg) synthesis in the fat body, secretion into hemolymphs, transport through intercellular channels in follicular epithelium, endocytosis mediated by Vg receptor (VgR), and absorption and storage by mature oocytes. In two representative Coleoptera species, *Leptinotarsa decemlineata* (Chrysomelidae) and *Henosepilachna vigintioctopunctata* (Coccinellidae), we performed RNA interference targeting ecdysone receptor (EcR) or ultraspiracle (usp) genes. Depletion of the expression level of *EcR* or *usp* inhibited oocyte development and dramatically repressed the transcription of *Vg* in fat bodies. Our findings indicate that 20E signaling plays an indispensable role in the stimulation of Vg synthesis and uptake in the two beetle species.

**Abstract:**

Coleoptera is the largest taxa of animals by far. The robust reproductive capacity is one of the main reasons for such domination. Successful female reproduction partially relies on effective vitellogenesis. However, the hormone regulation of vitellogenesis remains to be explored. In the present paper, in vitro culture of *Leptinotarsa decemlineata* 1-day-old adult fat bodies in the 20E-contained median did not activate juvenile hormone production and insulin-like peptide pathways, but significantly stimulated the expression of two *LdVg* genes, in a cycloheximide-dependent pattern. In vivo RNA interference (RNAi) of either ecdysone receptor (*LdEcR*) or ultraspiracle (*Ldusp*) by injection of corresponding dsRNA into 1-day-old female adults inhibited oocyte development, dramatically repressed the transcription of *LdVg* genes in fat bodies and of *LdVgR* in ovaries; application of JH into the *LdEcR* or *Ldusp* RNAi *L. decemlineata* females did not restore the oocyte development, partially rescued the decreased *LdVg* mRNA levels but over-compensated *LdVgR* expression levels. The same RNAi experiments were performed in another Coleoptera species, *Henosepilachna vigintioctopunctata*. Little yolk substances were seen in the misshapen oocytes in the *HvEcR* or *Hvusp* RNAi ovaries, in contrast to larger amounts of yolk granules in the normal oocytes. Correspondingly, the transcript levels of *HvVg* in the fat bodies and ovaries decreased significantly in the *HvEcR* and *Hvusp* RNAi samples. Our results here show that 20E signaling is indispensable in the activation of vitellogenesis in the developing oocytes of the two beetle species.

## 1. Introduction

Coleoptera is currently the most species-rich group, including about one fourth of all known insects with more than 360,000 described species on this planet. An enormous reproductive capacity is one of the main reasons for such domination [1]. To some extent, successful female reproduction depends on functioning vitellogenesis, a process that leads to the accumulation of vitellogenin (Vg) in the oocytes [2,3]. Insect Vgs are mainly produced in the fat body [3,4]. In some insect species, Vgs have also been reported to be synthesized in ovarian tissues [5], follicle cells [6], nurse cells [7], and haemocytes [8]. The Vg proteins are transported to the ovary through circulating hemolymph, across the patency among the follicle epithelium cells that enclose oocytes, absorbed through Vg receptor (VgR), and stored by oocytes [3,8].

Cumulative studies have established that ecdysteroids (the most active form, 20-hydroxyecdysone, 20E) and juvenile hormones (JH) regulate the synthesis and absorption of Vgs [3,9,10]. In Coleoptera, however, the reproduction regulation in the female is complex and less explored. JH has even been documented regulating vitellogenesis in some beetles [9,11]. For example, a positive correlation between JH synthesis and ovarian development has been reported in *Leptinotarsa decemlineata* (Chrysomelidae) [12], *Tribolium castaneum* (Tenebrionidae) [11], *Coccinella septempunctata* (Coccinellidae) [13], and *Anthonomus grandis* (Curculionidae) [14]. In *T. castaneum*, application of JH III can induce *TmVg* expression in the previtellogenic females. Conversely, suppressing JH signal impairs *Vg* gene expression and Vg protein accumulation [11]. Further research displays that the JH signaling pathway promotes insulin-like peptide 2 (ILP2) production through Met. As a result, the expression of the *Vg* gene is activated. JH also indirectly regulates vitellogenesis by inducing ILP-IGF production, and ILP-IGF subsequently activates an ILP cascade [15]. Moreover, the Bursicon neuropeptide signal also induces *Vg* expression, possibly by mediating the expression of JH and ILP cascade genes [16]. These data demonstrate that JH is necessary for the production of Vg protein by the beetle fat body.

It has also been known that 20E signaling is associated with vitellogenesis in two Coleoptera, *T. castaneum* (Tenebrionidae) and *Colaphellus bowringi* (Chrysomelidae) [9,11,17]. In *T. castaneum*, RNA interference (RNAi) studies show that vitellogenesis and oocyte maturation requires 20E response genes, such as Ecdysone-induced protein 75 (*E75*), Ecdysone-induced protein 93F (*E93*), hormone receptor 3 (*HR3*), hormone receptor 4 (*HR4*), Ecdysone receptor (*EcR*), ultraspiracle (*USP*), and βFTZ transcription factor 1 (β*FTZ-F1*) [18,19]. Moreover, Cap ‘n’collar isoform C, a transcription factor that activates the genes encoding for ecdysteroid biosynthesis enzymes, regulates the tradeoff between detoxification and reproduction [20]. In *C. bowringi*, exogenous application of either 20E or ecdysis-triggering hormone peptide (ETH) induces vitellogenesis. Any 20E deficiency not only decreases ETH level, but also reduces JH production, while knockdown of the *ETH* gene decreases the expression of *Vg1*, *Vg2*, and a few JH biosynthetic genes [17]. The mitogen-activated protein kinase signaling pathway possibly regulates ecdysone biosynthesis; disruption of this pathway delays ovarian development and causes low fecundity [21]. Similarly, knockdown of the coat protein II complex gene represses the ecdysone signaling pathway. As a result, yolk deposition and ovarian growth are considerably inhibited [22].

In *T. castaneum* and *C. bowringi*, 20E signaling triggers JH and/or ILP signaling to activate vitellogenesis. Whether the same hormonal regulative mode is conserved in other Coleoptera species remains to be determined. In order to compare the similarities within a family and the differences between the families in terms of oogenesis, in the present paper we selected two Coleoptera potato pests, *L. decemlineata* (Chrysomelidae) and *Henosepilachna vigintioctopunctata* (Coccinellidae). We intended to determine the role of 20E signaling in the regulation of vitellogenesis and tried to address two long-standing issues. (1). Is the 20E cascade necessary for the activation of vitellogenesis? (2). Is the gonadotrophic role of 20E signaling independent of the JH cascade? Our results here suggest the importance of 20E signaling in the vitellogenesis in the two Coleoptera pests.

## 2. Materials and Methods

### 2.1. Insect Rearing

*H. vigintioctopunctata* beetles were collected on *Solanum melongena* L. during the summer in Nanjing (24°32′00.00″ N; 117°22′00.00″ E), Jiangsu Province in China. The beetles were cultured in a laboratory at 28 ± 1 °C, 16 h:8 h light–dark photoperiod and under 50–60% relative humidity conditions, using fresh potato leaves. Under such feeding conditions, the duration from egg-laying to adult eclosion took about 20 days.

*L. decemlineata* adults were collected in spring, on the potato fields of Urumqi (43°82′55″ N, 87°61′68″ E), Xinjiang Uygur autonomous region in China. The insects were routinely fed on fresh potato leaves in an insectary at 28 ± 1 °C, 14 h:10 h light–dark photoperiod and 50–60% relative humidity. Under such feeding conditions, the duration from egg to adults was around 28 days.

### 2.2. Molecular Cloning

The *Henosepilachna vigintioctopunctata* and *Leptinotarsa decemlineata EcR* isoforms were downloaded from the National Center for Biotechnology Information (NCBI) (accession numbers: *HvEcRA*, BAP15927.1; *HvEcRB1*, BAP15926.1; *LdEcRA*, AB211191; *LdEcRB1*, AB211192). The *usp* isoforms were obtained from NCBI (accession numbers: *Hvusp1*, AB506671.1; *Hvusp2*, AB506672.1; *Ldusp1*, MH492017; *Ldusp2*, MH492018). Polymerase chain reaction (PCR) was performed using the primers in Appendix A to verify the sequence correctness of these target genes.

### 2.3. Synthesis of dsRNAs

Using the online siRNA design website (http://sidirect2.rnai.jp/) URL (accessed on 22 April 2022), highly efficient and specific siRNA fragments originating from the common *HvEcR* or *Hvusp* fragments or *HvEcRB1* isoform-specific sequence were selected. Then, primer premier 5.0 software was used to design a pair of primers (Appendix A) to amplify cDNA containing siRNA fragments. Further BLASTN searches of target cDNA sequences with *H. vigintioctopunctata* transcriptome data [23] were performed to determine any off-target sequences that might have an identical match of 20 bp or more. For *L. decemlineata*, primer pairs for *LdEcRA*, *LdEcRB1*, and *Ldusp* (ds*Ldusp-1*, ds*Ldusp-2*) are listed in Appendix A. In addition, a cDNA fragment of enhanced green fluorescent protein (egfp) gene was obtained from *Aequorea victoria*. PCR amplification was performed with specific primers conjugated with the T7 RNA polymerase promoter sequence (Appendix A). According to the instructions provided by the kit, dsRNA was synthesized in vitro using a MEGAscript T7 high-yield transcription kit (Ambion, Austin, TX, USA). The quality of synthetized dsRNA was detected by agarose gel electrophoresis, and the concentration (5–8 μg/μL) of the synthetized dsRNA was measured by a NanoDrop1000 spectrophotometer. The resultant dsRNA products were stored at −80 °C before experiments.

### 2.4. Injection of dsRNAs and Bioassay

Injecting dsRNA was performed according to a documented method [24,25]. In short, 0.1 μL of solution containing 500 ng dsRNA was injected into the newly emerged female adults. Negative control adults were injected with the same volume of ds*egfp* solution.

Juvenile hormone III (JH) (purity > 65%) and ecdysteroid 20-hydroxyecdysone (20E) (purity > 93%) were purchased from Sigma-Aldrich (St. Louis, MO, USA). JH was further purified by reverse-phase high-performance liquid chromatography (RP-HPLC); its final purity reached 90%. 20E and JH were dissolved in distilled water containing surfactant (Tween 20, 1 g/L) to obtain, respectively, 10^−3^ M and 100 ng/mL stock solutions.

For *L. decemlineata*, a total of 900 newly emerged female adults were used to carry out two independent biological experiments. A replicate contains 10 individuals, and a total of 90 replicates were set. The first was intended to determine the function of *Ldusp*; 450 newly emerged female adults were randomly separated into 45 replicates (groups); 9 groups were injected with ds*egfp*, 18 replicates with ds*Ldusp-1* and another 18 groups with ds*Ldusp-2*. Nine ds*egfp-*, 9 ds*Ldusp-1-* and 9 ds*Ldusp-2-*treated groups were fed on normal potato leaves. Nine ds*Ldusp-1-* and 9 ds*Ldusp-2-*treated groups were fed on potato leaves immersed with 30 mL JH (100 ng/mL). The second experiment was intended to evaluate the function of *LdEcR*; a total of 450 newly emerged female adults were randomly separated into 45 replicates; 9 groups were injected with ds*egfp*, 18 groups with ds*LdEcRA* and 18 groups with ds*LdEcRB1*. Nine ds*egfp-*, 9 ds*LdEcRA-* and 9 ds*LdEcRB1-*treated female beetles were fed on normal potato leaves. Nine ds*LdEcRA-* and 9 ds*LdEcRB1-*treated groups were fed on potato leaves immersed with 30 mL JH (100 ng/mL). One day later, these female adults were fed with fresh potato foliage. Three days after injection, three repeated samples were taken for qRT-PCR to detect the efficacy of RNA interference. Ten days after injection, three repeats were used to measure the expression levels of vitellogenesis genes (*LdVg1*, *LdVg2* and *LdVgR*). Six days after injection, another three replicates were used for dissection for observation under a microscope.

For *H. vigintioctopunctata*, three independent biological experiments were carried out using the newly emerged female adults, with two treatments: (1) ds*egfp* and (2) ds*HvEcR,* ds*HvEcRB1* or ds*Hvusp*. A group of 10 injected newly emerged female adults was set as a replicate. For ds*HvEcR* RNAi experiment, each treatment had 21 replicates. A total of nine replicates were harvested 3, 10 and 20 days after treatment. Three days after injection, three repeated samples were taken for qRT-PCR to detect the efficacy of RNAi. Ten and twenty days after treatment, the other six replicates were used to measure the expression levels of vitellogenesis genes (*HvVg* and *HvVgR*). Another 12 repeated treatments were collected 10, 15 or 30 days after injection and used to dissect and observe the ovaries under a microscope or used for sectioning and HE staining. For ds*HvRcRB1* RNAi experiment, each treatment had nine replicates. Three and ten days after treatment, a total of six replicates were harvested to extract total RNA. The other three repeats were used to observe the 10-day-old ovaries after dissection. For ds*Hvusp* RNAi experiment, each treatment had 21 replicates. A total of nine replicates were harvested 3, 10 and 20 days after treatment. Three days after injection, three repeated samples were taken for qRT-PCR to detect the efficacy of RNA interference. Ten and twenty days after treatment, the other six replicates were used to measure the expression levels of vitellogenesis genes (*HvVg* and *HvVgR*). Another 12 repeated samples were collected and dissected 10, 20 and 30 days after injection, observed under microscope or stained with HE.

### 2.5. In Vitro Fat Body Culture

One-day-old *L. decemlineata* adult fat bodies were dissected in saline. The isolated fat bodies were washed with culture medium four times and then cultured independently in 25 °C EX-CELL^®^ 405 Serum-Free Medium for Insect Cells (Sigma-Aldrich, USA). The fat bodies from 12 individuals were cultured in culture medium (control), 10^−4^ M cycloheximide (Chx) (Sigma-Aldrich, USA), 10^−6^ M 20E, or 10^−4^ M Chx + 10^−6^ M 20E. The treatment was repeated three times. Samples were randomly selected at 0, 3, 6, 12 and 24 h after incubation for mRNA expression analysis.

### 2.6. Quantitative Real-Time PCR

For analysis of the effects of treatments, total RNA was extracted from treated adults. Each sample contained 10 individuals and were repeated three times. The collected RNA samples were extracted using SV Total RNA Isolation system Kit (Promega, Madison, WI, USA), and DNase I was used to remove any residual DNA from the purified RNA. For *H. vigintioctopunctata*, using two internal control genes (*HvRPS18* and *HvRPL13*, primers listed in Appendix A) according to the published results [26]. For *L. decemlineata*, using four internal control genes (*LdRP4*, *LdRP18*, *LdARF1* and *LdARF4*, the primers listed in Appendix A) according to our published results [27]. Quantitative measurements of mRNA by qRT-PCR in a triplicate technique. The 2^−ΔΔCT^ method was used for calculating the relative value, using the geometric mean of internal control genes for normalization.

### 2.7. Hematoxylin-Eosin (HE) Staining

The phenotypic defects of ovaries were observed by HE staining. Simply, 10-, 20- and 30-day-old ovaries were dissected after the adults were treated with ds*egfp*-, ds*EcR* and ds*usp*, and were then fixed with 4% paraformaldehyde and embedded in paraffin. Then, the paraffin-embedded ovarian tissue was cut into 6-μm sections. Ovarian tissue sections were hydrated and stained with Mayer’s H&E (Yeasen, Shanghai, China), and then observed under an Olympus BH2 light microscope (Olympus, Tokyo, Japan).

### 2.8. Data Analysis

SPSS for Windows (Chicago, IL, USA) were used for statistical analysis. After ensuring the normal distribution of the data, multiple comparisons were made by one-way analysis of variance (ANOVA) and the Tukey–Kramer post-test. Some data were compared by unpaired Student’s *t*-test. Values (mean ± SE) of *p* < 0.05 were regarded as significant.

## 3. Results

### 3.1. In Vitro Induction of Hormonal and Vitellogenesis Genes by 20E in L. decemlineata

The fat bodies of newly emerged female adults were dissected and cultured with 20E/cycloheximide (Chx) (Figure 1). In order to determine whether the incubation in the presence of 10^−6^ M 20E activates 20E signaling, the expression of late ecdysone response gene *LdFTZ-F1* was measured. As expected, an addition of 20E greatly induced *LdFTZ-F1* transcription 3, 6, 12 and 24 h after treatment. However, an addition of 10^−4^ M Chx (the protein synthesis inhibitor) in the 20E-containing medium almost completely abrogated the induction. It appears that the stimulation of 20E signaling needs protein synthesis (Figure 1A).

Similarly, the incubation in the presence of 10^−6^ M 20E activated the expression of *LdVg1* and *2* in the fat bodies. The activation of transcription of *LdVg1* and *2* was also inhibited by Chx (Figure 1B,C).

In contrast, the incubation in the presence of 20E or 20E + Chx could not up-regulate the expression of *LdVgR*, *Krüppel homolog 1* (*LdKr-h1*) and Insulin receptor (*LdInR1)* (Figure 1D–F).

### 3.2. RNAi of usp Represses Vitellogenesis in L. decemlineata

Introduction of two dsRNAs (ds*Ldusp-1* and ds*Ldusp-2*) targeting different regions of *Ldusp* into the 1-day-old female adults reduced the expression levels to similar extents (Figure 2A).

The 6-day-old ovaries were dissected and their structures were observed. In the control ovarioles, the absorption of the yolk was initiated but not completed. The yellow-colored yolk molecules were accumulated in the developing oocytes (Figure 2B). The accumulation of yellow-colored yolk substances was hindered in the *Ldusp* RNAi beetles. The hindrance could not be relieved by the introduction of JH (Figure 2C,D).

The transcript levels of genes involved in the vitellogenesis were measured 10 days after dsRNA injection. The expression levels of both *LdVg1* and *LdVg2* were reduced in the *Ldusp* RNAi fat bodies. The reduction could be partially restored by JH application (Figure 2E,F).

For vitellogenin receptor gene *LdVgR*, its transcript levels declined in the *Ldusp* knockdown ovaries, which could be over-rescued by JH introduction (Figure 2J).

### 3.3. RNAi of EcR Isoforms Mirrors the Inhibition of Vitellogenesis in L. decemlineata

Isoform-specific dsRNA (ds*LdEcRA*, ds*LdEcRB1*) [24] were individually introduced into the 1-day-old adult females. In the females having suffered from a ds*LdEcRA* injection, *LdEcRA* + *LdEcRB1* (hereafter *LdEcR*) and *LdEcRA* mRNA levels were significantly decreased. In beetles injected with ds*LdEcRB1*, transcription levels of *LdEcR* and *LdEcRB1* were significantly reduced, while expression levels of *LdEcRA* were significantly increased (Figure 3A–C). Treatment with JH did not significantly change the expression levels of *LdEcRA*, *LdEcRB1*, or *LdEcR* in the treated females (Figure 3A–C).

Knockdown of either *LdEcRA* or *LdEcRB1* delayed the accumulation of yolk; the ovarioles appeared to be almost transparent (Figure 3E,G). Introducing JH into the *LdEcRA* or *LdEcRB1*RNAi females did not rescue the delay of the yolk accumulation, estimating by the almost transparent ovarioles (Figure 3F,H).

In the fat bodies, knockdown of either *LdEcRA* or *LdEcRB1* significantly reduced the mRNA levels of both vitellogenin genes *LdVg1* and *LdVg2*. Introduction of JH into the *LdEcRA* or *LdEcRB1* RNAi females partially alleviated the reduction of the mRNA contents (Figure 3I,J). In the ovaries, RNAi of *LdEcRA* or *LdEcRB1* did not reduce the expression levels of *LdVg1* or *LdVg2*, whereas application of JH stimulated the transcription of *LdVg1* (Figure 3L,M).

Consistently, *LdVgR* levels declined in the *LdEcRA* and *LdEcRB1* RNAi ovaries. JH application over-compensated *LdVgR* expression levels (Figure 2N). In contrast, the levels of *LdVgR* were higher in the ds*LdEcRA*, ds*LdEcRB1*, ds*LdEcRA* + JH, and *dsLdEcRB1* + JH-treated fat bodies than that in the control sample (Figure 3K).

### 3.4. Depletion of HvEcR Impairs Vitellogenesis in H. vigintioctopunctata

After knockdown of both *HvEcR* isoforms (Figure 4A), the phenotypic defects of the resultant females were examined.

In the normal eggs, the yellow-colored yolk substances were evenly distributed (Figure 4B,D). Conversely, the *HvEcR* RNAi eggs were misshapen and light-colored; the yellow matters were irregularly dispersed, with some colorless patches here and there (Figure 4C,E).

HE staining revealed that the normal eggs contained large amounts of yolk granules when examined 10, 15 and 30 days after eclosion (Figure 4F,H,J). Conversely, little yolk was accumulated in the 10-day-old *HvEcR* RNAi eggs (Figure 4G). Although some yolk substances were seen in the 15- and 30-day-old *HvEcR* RNAi eggs, they did not form granules (Figure 4I,K).

In order to determine the effects of *HvEcR* depletion on vitellogenin accumulation, the expression levels of vitellogenin and vitellogenin receptor genes *HvVg* and *HvVgR* in the fat body and ovary were compared at the age of 10 (Figure 4L–O) and 20 (Figure 4P–S) days. Knockdown of *HvEcR* significantly decreased the expression levels of *HvVg* in both the fat body and ovary (Figure 4L,N,P,R). The expression of *HvVgR* was complex: its levels were diminished in 10-day-old fat bodies and 20-day-old ovaries (Figure 4M,S), but were increased in 20-day-old fat bodies and 10-day-old ovaries (Figure 4O,Q) in the *HvEcR* knockdown samples, compared with the ds*egfp*-treated group.

### 3.5. RNAi of HvEcRB1 Delays Vitellogenesis in H. vigintioctopunctata

As the common region of *HvEcRA* and *HvEcRB1* sequences was long, and no effective target fragment was predicted in the *HvEcRA* isoform, we designed a dsRNA to silence the *HvEcRB1* (ds*HvEcRB1*) sequence.

Three days after ds*HvEcRB1* injection, the levels of *HvEcRB1* and the total level of two isoforms (*HvEcR*) were significantly reduced (Figure 5A,B). In contrast, the mRNA level of *HvEcRA* was not changed (Figure 5C). The expression of six 20E signaling genes (*HvE74*, *HvE75*, *HvE93*, *HvHR3*, *HvHR4* and *HvFTZ*-F1) was also measured. Knockdown of *HvEcRB1* did not decrease the mRNA levels of these genes (Appendix A).

The transcript levels of *HvVg* in the fat body and *HvVgR* in the ovary were detected. Depletion of *HvEcRB1* isoform negatively affected the expression of the two genes (Figure 5D,E). Consistently, the *HvEcRB1* RNAi ovary contained less mature eggs (Figure 5H vs. Figure 5G); the 10-day-old *HvEcRB1* RNAi females laid fewer eggs (Figure 5F).

### 3.6. Silence of Hvusp Disrupts Vitellogenesis in H. vigintioctopunctata

Injection of ds*Hvusp* significantly reduced *Hvusp* mRNA level when measured three days after treatment (Figure 6A). In the 10-day-old normal ovaries, a matured oocyte was seen in each ovariole. Conversely, the primary oocytes were not matured in the 10-day-old *Hvusp* RNAi ovarioles (Figure 6B vs. Figure 6C). In the 30-day-old *Hvusp* RNAi ovarioles, misshapen eggs were accumulated; the colored matters were irregularly dispersed (Figure 6D).

HE staining revealed that the normal eggs contained large amount of yolk granules when examined 10 and 20 days after eclosion (Figure 6E,G). Conversely, little yolk was accumulated in the 10- and 20-day-old *Hvusp* RNAi eggs (Figure 6F,H).

The expression levels of *HvVg* and *HvVgR* in the fat body and ovary were detected at the age of 10 (Figure 6I–L) and 20 (Figure 6M–P) days. RNAi of *Hvusp* significantly decreased the expression levels of *HvVg* in both the fat body and ovary (Figure 6I,K,M,O). Similarly, the *HvVgR* levels were significantly diminished in 10-day-old fat bodies and ovaries (Figure 6J,L), but were greatly increased in 20-day-old fat bodies (Figure 6N) in the *Hvusp* knockdown samples, compared with the ds*egfp*-treated group.

## 4. Discussion

Vitellogenesis is characterized by the synthesis of Vg in the fat body and secretion into the hemolymph, which is transported to the oocyte through the intercellular channels (patency) in the follicular epithelium and is absorbed and stored by the maturing oocyte [28]. The regulation of Vg synthesis, patency formation and Vg uptake are three key steps to affect Vg deposition in oocytes. In the current article, we examined the requirements of 20E signaling for vitellogenesis. Our functional analysis shows that 20E signaling plays a key role in the regulation of Vg synthesis and uptake in two representative Coleoptera species.

### 4.1. 20E Signaling Regulates Vg Synthesis in Fat Body, in a JH-Independent Pattern

In the current paper, we discovered that in vitro culture of *L. decemlineata* 1-day-old fat bodies with 20E significantly stimulated the expression of both *Vg* genes but not *Kr-h1* and *InR1* genes, in a cycloheximide-dependent pattern (Figure 1). Our findings indicate that 20E signaling activates *Vg* biosynthesis in a JH and ILP signals-independent manner. In line with our result, cumulative studies have confirmed that vitellogenesis is activated by the rise of ecdysteroid titer in some Hymenoptera, Lepidoptera, Diptera and Coleoptera insect species [29,30,31]. In *Drosophila melanogaster*, for instance, the presence of 20E contributes to the high Vg synthesis rate in the fat body [32]. Similarly, in *Aedes aegypti*, 20E stimulates *Vg* expression in the fat body and oocyte maturation in the ovary after a blood meal [33]. In *T. castaneum*, knockdown of 20E signaling pathway genes blocks Vg synthesis [4].

Repression of 20E-stimulated *Vg* expression by cycloheximide addition (Figure 1) demonstrates that protein synthesis is involved. It implies that active 20E cascade is critical for the stimulation of Vg biosynthesis in the fat body in *L. decemlineata*. Therefore, we knocked down either *EcR* or *usp* in the two Coleoptera species. Our results uncovered that the knockdown dramatically suppressed the transcription of *Vg* in adult fat bodies of the two beetle species (Figure 2, Figure 3, Figure 4 and Figure 6). Reduced *Vg* mRNA levels may lower the corresponding protein contents. As a result, oocyte maturation would be delayed in both beetle species. Thus, we observed that RNAi of *EcR* or *usp* in the two Coleoptera species delayed yolk accumulation and oocyte growth (Figure 2, Figure 3, Figure 4, Figure 5 and Figure 6). Application of JH into the *EcR* or *usp* RNAi *L. decemlineata* females only partially rescued the decreased *Vg* levels (Figure 2 and Figure 3). The in vivo RNAi results provided another piece of evidence that 20E signaling regulates Vg synthesis in the fat body, in a JH-independent pattern.

Consistently, in some Lepidopterans such as *Bombyx mori*, *Hyalophora cecropia*, and *Spodoptera frugiperda*, 20E has a primary role in vitellogenesis [29,34]. Repression of 20E synthesis remarkably reduces the expression levels of *Vg* and *VgR*, resulting in egg development defects of *Plutella xylostella* [29]. A similar stimulatory role of 20E signaling in Vg synthesis has also been documented in *D. melanogaster* [13].

### 4.2. Interaction of 20E and JH Signaling Enhances Vg Synthesis in Fat Body

Partial rescuing of *Vg* expression by JH in the *EcR* or *usp* RNAi females in *L. decemlineata* (Figure 2 and Figure 3) also suggests interaction of 20E and JH signaling enhances Vg synthesis in fat body.

In agreement with our result, the interactions of 20E, JH and ILP cascades during stimulation of Vg biosynthesis have been documented in two beetle species, *T. castaneum* and *C. bowringi* [9,11,17], and non-Coleoptera insect species [2,35]. Firstly, 20E signaling, through the downstream component ETH, stimulates JH biosynthesis. In *Periplaneta americana*, injection of 20E inhibits the transcription of juvenile hormone acidmethyltransferase (*jhamt*), *Vg* and *VgR* [36]. In *D. melanogaster*, ETH plays a crucial role in the maintenance of JHAMT [37]. In *Bactrocera dorsalis*, injection of dsRNA targeting ETH or ETH receptor (ETHR) into female adults reduces the expression of *jhamt*, *Vg2*, and a few JH signal genes; 20E or methoprene can restore egg production to normal levels [35]. Secondly, JH signaling enhances the 20E cascade. For example, in *D. melanogaster*, JH acts on the phosphorylation of USP through the receptor tyrosine kinase-phospholipase C-protein kinase C (RTK-PLC-PKC) cascade, thus enhancing the role of 20E in activating Vg production [38]. In *A. aegypti*, JH promotes the ability of fat body for Vg synthesis, while 20E stimulates *Vg* expression in the fat body and oocyte maturation in the ovary after a blood meal [33]. Thirdly, 20E signaling can interact with the ILP pathway. For example, in *A. aegypti*, ETH mobilizes calcium ions from the endoplasmic reticulum through IP3 receptors to regulate the activities of JHAMT and JH [13]. Meanwhile, insulin and bombyxin (an insulin-like hormone) are directly stimulated in the prothoracic glands in *B. mori*. Both insulin and prothoracicotropic hormone (PTTH) stimulate the ecdysteroid secretion by increasing the phosphorylation of Akt [39].

How 20E signaling interacts with JH cascade during vitellogenesis in the two Coleoptera species in this deserves further investigation.

### 4.3. Isoform Specific Role of EcR in the Regulation of Vg Accumulation

In *L. decemlineata*, knockdown of either isoform of *EcR* suppressed Vg accumulation, indicated by the decreased *Vg* levels and the transparent ovarioles in the resultant females (Figure 3). In *H. vigintioctopunctata*, RNAi of *HvEcRB1* greatly reduced the expression of *HvVg*. Moreover, the *HvEcRB1* RNAi ovary contained less mature eggs and the 10-day-old *HvEcRB1* RNAi females laid fewer eggs (Figure 5).

Consistently, EcR isoform-dependent regulation of oogenesis has been documented in other insects. In *D. melanogaster*, 20E regulates follicle rupture and ovulation by activating *EcRB2*. The deletion of EcR reduces egg laying of females, which can be reversed by ectopic expression of *EcRB2* [40]. Moreover, the eggshell gene *VM32E* can produce components of the vitelline membrane and endochorion layers, and transcription of the gene is activated by EcRB1 and USP [41].

In this survey, we found that knockdown of *HvEcRB1* did not decrease the mRNA levels of six 20E signaling genes (*HvE74*, *HvE75*, *HvE93*, *HvHR3*, *HvHR4* and *HvFTZ*-F1) (Appendix A). It appears that the EcRB1/USP complex may directly trigger the expression of *HvVg* and *HvVgR* during vitellogenesis in *H. vigintioctopunctata*. In line with our data, the *Vg* 5′ regulatory region contains several EcR response elements (EcREs) in *A. aegypti*, providing evidence of direct control of this gene by EcR-USP [42,43].

Comparison of the negative effect on vitellogenesis in the *HvEcRB1* and *HvEcRA* + *HvEcRB1* RNAi ovaries indicates that the latter exhibits a severe defective phenotype in *H. vigintioctopunctata* (Figure 5 vs. Figure 4). It appears that the 20E-EcRA/USP complex may also be associated with the stimulation of vitellogenesis by an unknown signaling cascade. One possible way is a positive signal from the normal ovaries. It is known that a normal ovary releases an ovarian factor at a specific stage of development that stimulates the corpora allata of *Diploptera punctata* to synthesis JH [44]. We accordingly propose that failure to initiate Vg transcription in the fat bodies of *EcR* or *usp* RNAi beetles (this study) and *T. castaneum* [4] partially comes from an indirect effect caused by ovarian maturation retardation. Another possible way is negative feedback from accumulated Vg in hemolymph. In the present paper, we found that *VgR* was also expressed in the fat body (Figure 2, Figure 3, Figure 4, Figure 5 and Figure 6), and that Vg may serve as an autocrine molecule binding to VgR to regulate the expression of its gene in the fat body. In terms of the signaling role of Vg, some results have been reported in worker bees, where Vg plays a role in sensing fat body sugars and in gustatory perception [31,44].

### 4.4. Hormonal Signals Control VgR Expression on Oocytes

During reproductive development, the expression of *VgR* is regulated by hormones. However, the regulation of *VgR* expression is complex. JH, 20E, ILP/TOR signaling pathways and micro RNAs have been documented to be involved [3].

In this paper, RNAi of either *Ldusp* or *LdEcR* significantly inhibited the expression of *LdVgR* in the ovaries in *L. decemlineata*. Meanwhile, application of JH into the *Ldusp* or *LdEcR* RNAi *L. decemlineata* females over-compensated *LdVgR* expression levels (Figure 2 and Figure 3). In another Coleoptera, *C. bowringi*, JH enhances *VgR* expression by the Met-Kr-h1 pathway [45]. Similarly, the RNAi of *BmKr-h1* in *B. mori* reduces the deposition of vitellin and leads to partially transparent chorion [46]. Kr-h1 seems to trigger the expression of VgR in the ovaries. In adults, Kr-h1 is induced by 20E and is a key participant in the JH signaling pathway [47]. On one hand, the Met–Taiman complex binds the JH response element in the *Kr-h1* promoter and directly regulates its transcription [1,48]. On the other hand, in vivo analysis displays that Kr-h1 is significantly up-regulated by 20E signaling in *D. melanogaster* [49,50] and *B. mori* [47,51]; in vitro culture shows that *Kr-h1* is significantly up-regulated by 20E in *T. castaneum* TcA cells [52] and in *Helicoverpa armigera* larval epidermis [53]. Moreover, 20E significantly synergizes the *BmKr-h1* induction by JH analog on cultured *B. mori* larval and pupal epidermis and the NIAS-Bm-aff3 cell line [54]. We can accordingly propose that Kr-h1 may also govern the expression of *LdVgR* in the *L. decemlineata* ovary. Given that *LdKr-h1* may be independently activated by both 20E and JH cascades, suppression of 20E signaling in the *LdEcR* and *Ldsup* RNAi adults down-regulates *LdKr-h1* expression and thus reduces the *LdVgR* level, while application of JH into *LdEcR* and *Ldsup* RNAi adults up-regulates *LdKr-h1* expression and thus over-compensates the *LdVgR* level.

Consistent with our proposal, knockdown of either *HvEcR* or *Hvusp* caused a complicated influence on the expression of *HvVgR* in *H. vigintioctopunctata* ovaries (Figure 4 and Figure 6). Therefore, the transcription of *VgR* may also be regulated by both 20E and JH signaling through Kr-h1. Actually, the complex regulation of *VgR* expression has widely been documented in other insect species. For instance, in *P. americana,* the expression of *VgR* is suppressed by additional 20E treatment [36]. Inhibition of 20E synthesis significantly decreased the expression levels of *VgR*, resulting in the development of defective eggs in *P. xylostella* [29]. In *Solenopsis invicta*, JH significantly increased the expression of *VgR* in cultured ovaries [51]. Analogously, JH treatment induced the expression of *VgR* in *Nilaparvata lugens* adult females [55]. Further research will shed light on the hormonal control of *VgR* expression in both *L. decemlineata* and *H. vigintioctopunctata* ovaries.

Our data here also displayed that even though application of JH partially recovered the decreased *LdVg* expression and over-compensated *LdVgR* expression levels in the *LdEcR* or *Ldusp* RNAi females in *L. decemlineata*, it does not change the transparency of the ovaries (Figure 2 and Figure 3). These findings suggest that 20E signaling directly hinders oocyte maturation in *L. decemlineata*, similar to the results in another Coleoptera, *T. castaneum* [5]. Obviously, hindrance of oocyte maturation inhibits uptake of Vg proteins in the *LdEcR* or *Ldusp* RNAi females in *L. decemlineata*. This issue deserves further investigation.

## 5. Conclusions

The data in the current paper suggest that 20E signaling, through the EcR/USP complex, induces vitellogenesis in two Coleoptera species, in an isoform-dependent way. Our research provides a solid foundation for future studies to understand the molecular mechanisms of reproduction regulation in beetles.

## Figures and Tables

**Figure 1 biology-12-01284-f001:**
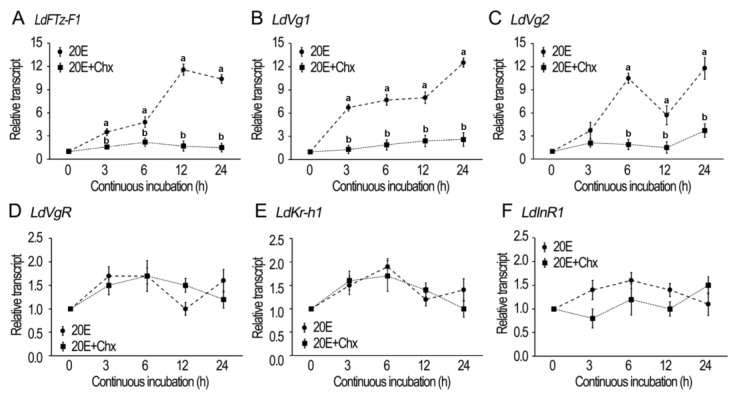
20E-induced expression of hormone response and vitellogenesis genes in *Leptinotarsa decemlineata*. The fat bodies from newly emerged female adults were incubated with a mixture of culture medium, 10^−4^ M cycloheximide (Chx), 10^−6^ M 20E, and 10^−4^ M Chx + 10^−6^ M 20E for 3, 6, 12, and 24 h. The expression levels of *LdFTZ-F1*, *LdVg1*, *LdVg2*, *LdVgR*, *LdKr-h1* and *LdInR1* genes were detected respectively (**A**–**F**). The expression level of the culture medium–incubation group was similar to that of the 10^−4^ M Chx-incubated group at each time point, and the average value was defined as 1. The dot represents the 2^−ΔΔCT^ method value (±SD), normalized to the geometrical mean of housekeeping gene expression. Different letters indicate significant difference at *p* value < 0.05.

**Figure 2 biology-12-01284-f002:**
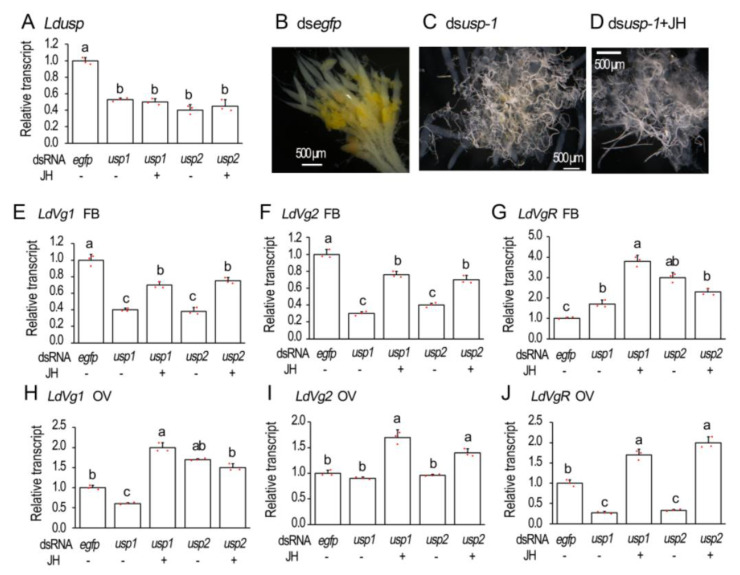
Knockdown of *Ldusp* represses vitellogenesis in *Leptinotarsa decemlineata*. The newly emerged female adults were treated with ds*egfp*, ds*Ldusp-1*, ds*Ldusp-2*, ds*usp-1* + 100 ng/mL JH and ds*Ldusp-2* + 100 ng/mL JH. The treated beetles were fed on fresh potato foliage. Three and ten days after treatment, transcript levels of *Ldusp* (**A**, 3d), *LdVg1* (**E**,**H**, 10d), *LdVg2* (**F**,**I**, 10d) and *LdVgR* (**G**,**J**, 10d) were determined. The relative transcripts refers to the ratios of relative copy numbers of the treated individuals to ds*egfp*-treated controls, which are set to 1. Different letters indicate significant difference at *p* < 0.05 using ANOVA with the Tukey–Kramer test. The ovaries of 6-day-old females were dissected and imaged (**B**–**D**).

**Figure 3 biology-12-01284-f003:**
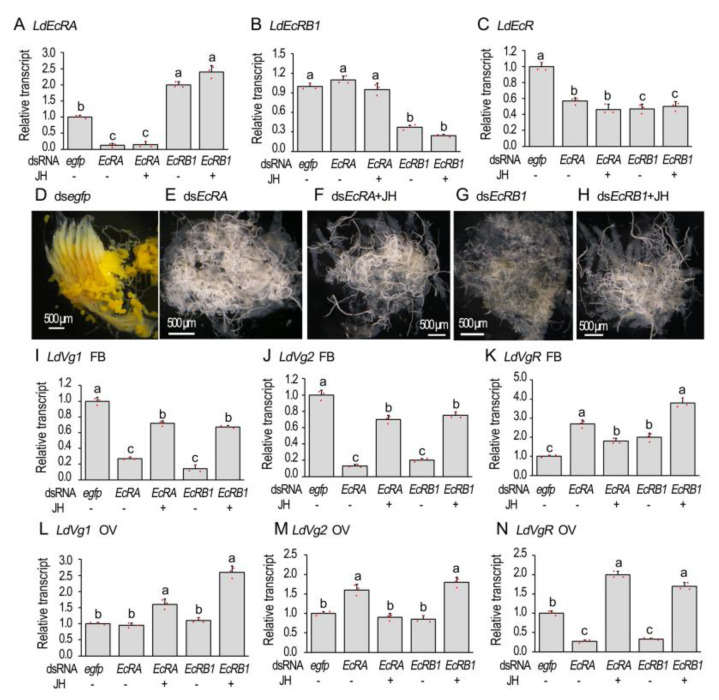
RANi of *LdEcR* inhibits vitellogenesis in *Leptinotarsa decemlineata*. The newly emerged female adults were treated with ds*egfp*, ds*LdEcRA*, ds*LdEcRB1*, ds*LdEcRA* + 100 ng/mL JH and ds*LdEcRB1* + 100 ng/mL JH. The treated beetles were fed on fresh potato foliage. Three and ten days after treatment, transcript levels of *LdEcRA*, *LdEcRB1* and both isoforms (**A**–**C**, 3d), *LdVg1* (**I**,**L**, 10d), *LdVg2* (**J**,**M**, 10d) and *LdVgR* (**K**,**N**, 10d) were determined. The relative transcripts refers to the ratios of relative copy numbers of the treated individuals to ds*egfp*-treated controls, which are set to 1. Different letters indicate significant differences at *p* < 0.05 using ANOVA with the Tukey–Kramer test. The ovaries of 6-day-old females were dissected and imaged (**D**–**H**).

**Figure 4 biology-12-01284-f004:**
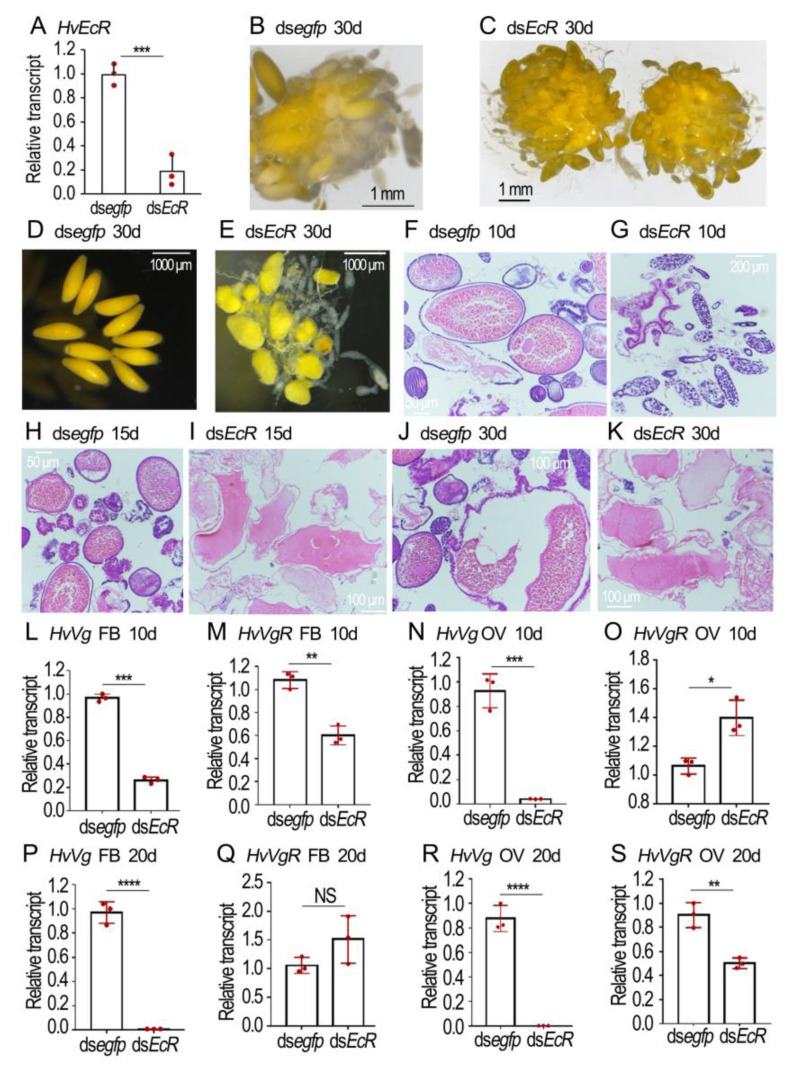
Depletion of *HvECR* impairs vitellogenesis in *Henosepilachna vigintioctopunctata*. The newly emerged female adults were treated with 0.1 μL ds*egfp* and ds*HvEcR* (400 ng) by injection. The treated beetles were fed on fresh potato foliage. Three, ten and twenty days (3d, 10d and 20d) after treatment, transcript levels of *HvEcR* (**A**), *HvVg* (**L**,**M**,**P**,**Q**) and *HvVgR* (**N**,**O**,**R**,**S**) were determined. The relative transcripts refers to the ratios of relative copy numbers of the treated individuals to ds*egfp*-treated controls, which are set to 1. Different instars indicate significant difference at *p* < 0.05 (*), 0.01 (**), 0.001 (***), or 0.0001 (****) using *t*-test. These columns use vertical lines to represent averages and indicating SD. NS, no significance. The ovaries of 30-day-old females were dissected and imaged (**B**–**E**). The ovaries from 10-, 15- and 30-day-old adults were sectioned, stained with hematoxylin-eosin staining (HE), and imaged (**F**–**K**).

**Figure 5 biology-12-01284-f005:**
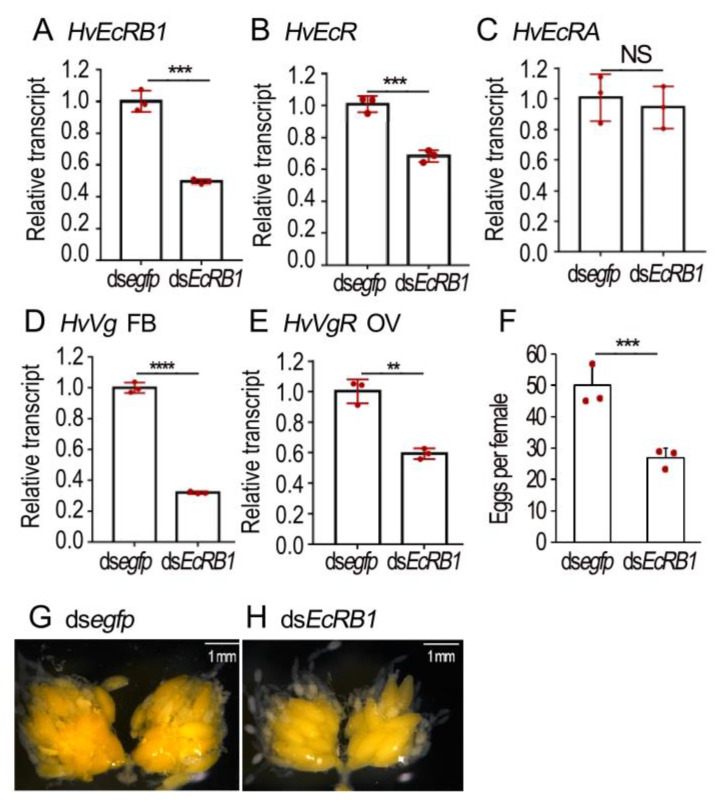
RNAi of *HvEcRB1* delays vitellogenesis in *Henosepilachna vigintioctopunctata* female adults. The newly emerged female adults were treated with 0.1 μL ds*egfp* and ds*EcRB1* (400 ng) by injection. The treated beetles were fed on fresh potato foliage. Three (**A**–**C**) or ten days (**D**,**E**) after treatment, transcript levels of *HvEcR* (two isoforms), *HvEcRB1* and *HvEcRA* in whole bodies, *HvVg* in fat body and *HvVgR* in ovaries were determined. The relative transcripts refers to the ratios of relative copy numbers of the treated individuals to ds*egfp*-treated controls, which are set to 1. The number of eggs was recorded in 10-day-old females (**F**). Different instars indicate significant difference at *p* < 0.01 (**), 0.001 (***), or 0.0001 (****) using *t*-test. These columns use vertical lines to represent averages and indicating SD. NS, no significance. The 10-day-old ovaries are shown (**G**,**H**).

**Figure 6 biology-12-01284-f006:**
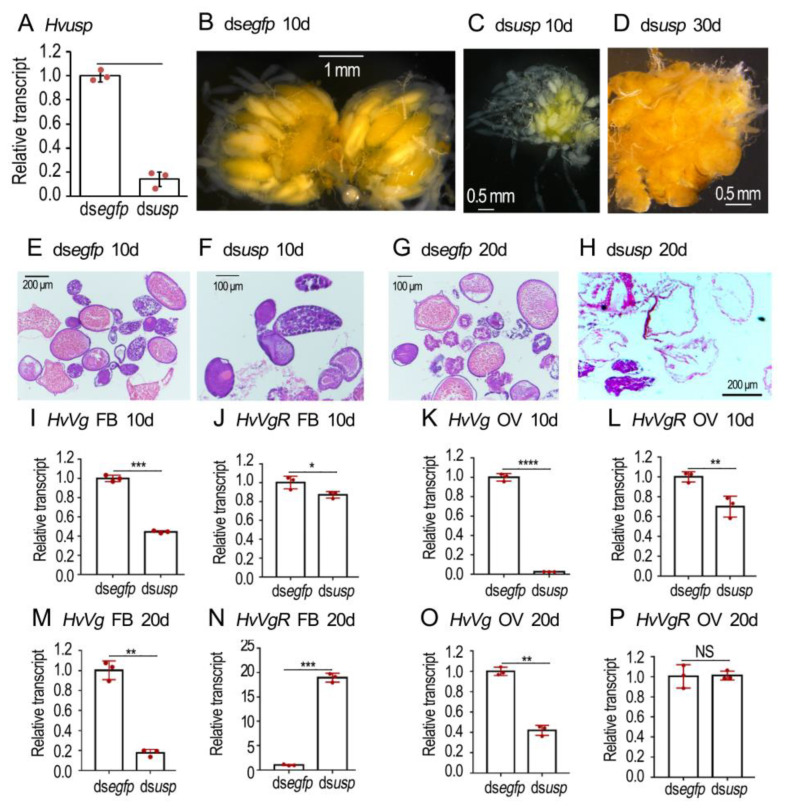
RNAi of *Hvusp* represses vitellogenesis in *Henosepilachna vigintioctopunctata*. The newly emerged female adults were treated with 0.1 μL ds*egfp* and ds*Hvusp* (400 ng) by injection. The treated beetles were fed on fresh potato foliage. Three, ten and twenty days (3d, 10d and 20d) after treatment, transcript levels of *HvEcR* (**A**), *HvVg* (**I**,**K**,**M**,**O**) and *HvVgR* (**J**,**L**,**N**,**P**) were determined. The relative transcripts refers to the ratios of relative copy numbers of the treated individuals to ds*egfp*-treated controls, which are set to 1. Different instars indicate significant difference at *p* < 0.05 (*), 0.01 (**), 0.001 (***), or 0.0001 (****) using *t*-test. These columns use vertical lines to represent averages and indicate SD. NS, no significance. The ovaries of 10- and 20-day-old females were dissected and imaged (**B**–**D**). The ovaries from 10- and 20-day-old adults were sectioned, stained with hematoxylin-eosin staining (HE), and imaged (**E**–**H**).

## Data Availability

Data generated in association with this study are available in the Appendix A published online with this article.

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
