# Peer review of "RNAi-Mediated Functional Analysis Reveals the Regulation of Oocyte Vitellogenesis by Ecdysone Signaling in Two Coleoptera Species"

_biology, 2023, doi:10.3390/biology12101284_

Round 1

Reviewer 1 Report

The manuscript describes studies that seek to characterize the role of ecdysteroid signaling in the regulation of vitellogenesis in two species of Coleoptera that are not closely related – one, a Chrysomelid and the other, Coccinelid – that are both pests of potatoes. It is unclear why the two are covered in the same manuscript, as no information is offered about the similarities or differences in their development, adult feeding/reproduction, or characteristics of potato infestation. This weakness also extends to the listing of coleopteran species (lines 55-57) for which there are studies of their endocrine regulation of oogenesis, but the authors do not link phylogenetic or biological relationships to the study species, which are critical considerations for the interpretation of experimental results given the singular speciation and diversity of Coleoptera. Furthermore, the manuscript is not easy to read or understand the interpretations of the results for the two species since the same experiments were not done in parallel for each species. It would be best to split the manuscript into separate ones for each species.

In general, there are many grammatical and structure errors that further erode my understanding of what is new or important that these studies contribute to our understanding of the endocrine regulation of reproduction in related groups of beetles, which should be the focus, and not the shotgun blast listings of what is similar or not to other unrelated model insect species found in the Discussion.

Additional suggestions or points for the authors to address:

Information about the source and titer profiles of ecdysteroids, JH, and peptides, such as insulin-like peptides and ETH in female adults of the species should be provided if known, as these parameters may differ and impact interpretation of experimental outcomes. Moreover, the authors should provide a summary figure for the sequence/function of ecdysteroid signaling elements and which were targeted in the adults used in the study – same for JH and insulin signal elements as mentioned.

The authors must know that target gene transcripts may go up or down in abundance in response to a treatment or RNAi, but those trends alone do not offer definitive evidence for effects on protein synthesis or abundance. Protein blots of tissues from experimental animals can be used to support such trends. For example: Ln 232/subheading 3.2. RNAi of usp represses vitellogenesis – no evidence is offered that Vg protein levels are affected, whereas it is clear that oogenesis is affected, based on the differing morphological evidence.

- Ln 66: VG or Vg – be consistent.

- Ln 69: Full terms and protein type need to be provided for ALL acronyms at first use.

- Ln 70: No genes “encoding for ecdysteroid biosynthesis” are known, but there are genes encoding for proteins and enzymes involved in this process.

- Ln 97 and 102: …to eclosion of adults…

- Ln 168: Does the medium contain a complete balance of amino acids? As this is known to be essential for activation of target of rapamycin kinase signaling, which enhances signaling through the other hormone pathways in insects.

There are many grammatical errors and excess content that need to be addressed.

Reviewer 2 Report

The MS “RNAi-mediated functional analysis reveals the regulation of oocyte vitellogenesis by ecdysone signaling in two Coleoptera species” tests the effect of RNA interference of ecdysone receptor and ultraspiracle genes on oocyte development and transcription of vitellogenin receptor gene in oosytes and vitellogenin genes in fat bodies of Leptinotarsa decemlineata and Henosepilachna vigintioctopunctata. The authors found out that the knockdown of both genes resulted in a vitellogenesis impairment in both L. decemlineata and H. vigintioctopunctata and JH application failed to rescue the oocyte development. The authors also demonstrated that 20E induced the expression of vitellogenin gene in L. decemlineata fat body in vitro. The conclusion concerning the obligatory role of 20E signal in the vitellogenesis induction through EcR/USP complex is made for two Coleoptera species under study.

The study is well setup, looks well executed and contains new data which are well discussed. The manuscript could be of interest to the readership of Biology.

Minor comments.

Line 17. Names of EcR and usp genes should be italicized.

Lines 56-58. Names of species should be italicized.

Lines 46-47. “In some cases, Vg has also synthesized been reported in

47 follicle cells [4], nurse cells [5], and haemocytes [2] in several insect species.”

I believe classic work by Jowett and Postlethwait should be mentioned here.

Jowett, T. & Postlethwait, J. H. The regulation of yolk polypeptide synthesis in Drosophila ovaries and fat body by 20-hydroxyecdysone and a juvenile hormone analog. Dev. Biol. 80, 225–234 (1980).

Reviewer 3 Report

The regulation of reproductive maturetion is a coplex system depending not only on ecdysteroids and juvenoids but on peptides and biogenic monoamines. The system must be turned on at right times and right situations. Although ILPs are examined, other essential factors such as other neuropeptides such as neuroparsin  and CCAP and biogenic monoamines are not examined. Therefore, it is incomplete. Starvation affects greatly but it has not been tested. The production of 20E occurs in the gonads and the ms fails to mention how the gland is first activated. The natural process should be sequentially described.

See the enclosed.

Round 2

Reviewer 1 Report

The authors satisfactorily addressed most of my suggestions and comments and improved the written English in the revised manuscript. The points below were not adequately answered and if done, would provide more depth to the interpretation and significance of the results. Other details should be cleared up as well.

1. Still no information is offered about the two coleopteran species as to similarities or differences in their development, adult feeding/reproduction, or characteristics of potato infestation. Since they are a chrysomelid and coccinellid species, then the family of the other coleopteran species (lines 55-57) should be given to link phylogenetic or biological relationships to the study species, which are critical considerations for the interpretation of experimental results given the singular speciation and diversity of Coleoptera.

2. Information about the commercial source and preparation of JH was given, but nothing was provided about the cellular sources and titer profiles of ecdysteroids, JH, and peptides, such as insulin-like peptides and ETH in female adults of the species or other coleopteran models. This information should be provided in the Introduction or Discussion if known, as these parameters may impact interpretation of experimental outcomes.

3. The author’s response to my point that the same experiments were not done in parallel for each species is nonsense: … “the same scientific method may cause similar systemic error even if it is used in different insect species, the best way to avoid the systemic error is using different method in various species to get the same results.” Numerous comparative studies of insects and other animals use the same experimental design/methods to aid comparison/contrast of the results.

4. Ln 168: Does the medium contain a complete balance of amino acids? As this is known to be essential for activation of target of rapamycin kinase signaling, which enhances signaling through the other hormone pathways in insects.

Yes. The medium used to culture fat bodies contain a complete balance of nutrients.

Specifics about the amino acids and other nutrients in the medium if known should be added in the Methods in regards to this signaling pathway.

5. Line 386 – 387: Hymenoptera etc. is capitalized NOT hymenopterans etc. That’s the convention and the rest of the manuscript needs to be checked for this recurring error.

6. Discussion and anywhere else: For every species mentioned, check to make sure the full scientific name is given at first mention along with the common names/orders so the reader will know the phylogenetic relationship/distance of the species to the coleopterans used in the study: e.g. lines 388-389: A. aegypti and D. melanogaster – check other lines below.

7. Lines 421 – 428: All acronyms (RTK…, PTTH) should be preceded by the full name/term of the protein/gene.

Minor corrections as suggested need to be made.
